# Development of a GIS-Based Methodology for the Management of Stone Pavements Using Low-Cost Sensors

**DOI:** 10.3390/s22176560

**Published:** 2022-08-31

**Authors:** Salvatore Bruno, Lorenzo Vita, Giuseppe Loprencipe

**Affiliations:** Department of Civil, Constructional and Environmental Engineering, Sapienza University, Via Eudossiana 18, 00184 Rome, Italy

**Keywords:** pavement monitoring, stone pavements, urban roads, international roughness index, pavement condition index, ride comfort, inertial sensor-based system, pavement management system, geographic information system

## Abstract

Stone pavements are present in many cities and their historical and cultural importance is well recognized. However, there are no standard monitoring methods for this type of pavement that allow road managers to define appropriate maintenance strategies. In this study, a novel method is proposed in order to monitor the road surface conditions of stone pavements in a quick and easy way. Field tests were carried out in an Italian historic center using accelerometer sensors mounted on both a car and a bicycle. A post-processing phase of that data defined the comfort perception of the road users in terms of the *a_wz_* index, as described in the ISO 2631 standard. The results derived from the dynamic surveys were also compared with the corresponding values of typical pavement indicators such as the International Roughness Index (IRI) and the Pavement Condition Index (PCI), measured only on a limited portion of the urban road network. The network’s implementation in a Geographic Information System (GIS) represents the surveys’ results in a graphical database. The specifications of the adopted method require that the network is divided into homogeneous sections, useful for measurement campaign planning, and adopted for the GIS’ outputs representation. The comparisons between IRI-*a_wz_* (R^2^ = 0.74) and PCI-*a_wz_* (R^2^ = 0.96) confirmed that the proposed method can be used reliably to assess the stone pavement conditions on the whole urban road network.

## 1. Introduction

Stone pavements represent a relevant architectural heritage in several Italian cities. It should be noted how the maintenance and management activities of these roads play a pivotal role not only in vehicle transit but also in the preservation of historical buildings.

At first, the main approach of many administrations was to replace historic stone pavements with modern asphalt pavements. However, in recent years there has been a revival of interest in both the restoration and construction of stone pavements [1,2,3,4]. In this regard, several studies [5,6,7,8] have shown that stone pavements reduce surface temperatures in urban areas. Nowadays, most of the stone paved roads are protected as they are considered cultural heritage like other ancient buildings, and they impact road traffic safety [9,10] and noise pollution [9,11,12,13]. Therefore, current management activities should include appropriate monitoring and maintenance techniques for stone pavements to ensure appropriate functional levels for users. The Pavement Management System (PMS) is the traditional tool for road agencies, characterized by a set of methodologies and procedures in order to plan proper maintenance activities according to the available budget and within a reference time [14,15,16,17,18]. Some researchers have shown that adopting a PMS to schedule preventive maintenance activities can significantly reduce costs for both users and administrations [19,20,21,22] compared with corrective maintenance without planning [23,24,25]. In recent years, the number of PMSs applied for urban systems has increased [2,26,27]. However, this trend has not involved stone pavements. Therefore, intervention thresholds values commonly provided by the current PMSs cannot also be adopted for stone pavements, pavement deterioration levels are generally defined in terms of pavement performance indicators (PPI), such as the International Roughness Index (IRI) and the Pavement Condition Index (PCI); it should be noted that IRI and PCI values for assessing stone pavement ride quality are a highly attractive problem [28,29]. In this regard, Zoccali et al. [26] proposed a new evaluation criterion based on the PCI method and defined IRI performance classes for stone pavements in some reference areas. The *a_wz_* index, as defined in the ISO2631 [30] standard, provides an alternative approach to evaluate pavement deterioration [31]. This comfort index can be calculated from the acceleration measured inside a test vehicle, by, for example, using the sensors embedded in modern smartphones [32,33,34,35].

Uploading data collected during field tests into a Geographic Information System (GIS) provides an easy way in order to quickly manage and visualize them, with also the opportunity to access additional data (e.g., historical data and maintenance works) where available [36]. The importance of this issue can be highlighted by some approaches already adopted. For example, some PMSs managed the data using hard-copy or digital maps and digital tables disconnected from spatial data [37]. In the 1980s, the Streetsaver software was developed by the Metropolitan Transportation Commission (MTC) of San Francisco [38]. It provides the catalogue of pavement distresses (as defined in the ASTM D6433 [39]) associated with the metropolitan transport graph; distress detection as an input enables the calculation and visualization of the PCI value for each section and branch. Further experimental analyses based on this methodology have made it possible to derive a performance deterioration curve for the PCI over time [40,41]. At the very beginning, GIS systems were employed by airport managers [14,42,43,44]; then, road authorities also included PMS data into GIS [22,25,45,46,47]. The implementation of a GIS in urban PMS has encouraged data collection and representation, as well as the management of different surveys and their representation [18,48]. Subsequently, several researchers [18,36,37,48,49,50,51] analyzed different case studies demonstrating the successful application of GIS in a PMS to optimize the data collection methodology and identify the best intervention and research allocation strategies. Thus, GIS offers a solution for maintaining a road database, which is very important for decision making and pavement management [48].

In this study, a novel method for quickly monitoring the road surface of stone pavements using accelerometer sensors mounted on a car and bicycle is discussed. The road surface detected is determined by the number of surveys, due to the well-known vehicles’ trajectory dispersion phenomena [52,53]: the proposed approach makes it easy to carry out multiple measurements for the investigated area, as the system is easy to install on several vehicles and does not require qualified personnel, and thus detect the entire lane.

The challenge and special feature of this study are also to investigate the potential of GIS systems applied on a stone road network characterized by different traffic conditions and control techniques.

Referring to a portion of the network investigated during the scheduled field tests, regression functions between different pavement performances (i.e., PCI, IRI, and *a_wz_*) have been defined. Next, the PPIs taken into account in this article are estimated using the *a_wz_* values calculated on the whole network and the regression functions. In addition, the regressions were implemented directly in GIS in order to derive a practical and fast system to identify the conditions of the investigated pavements with respect to traffic conditions. The proposed system allows qualified personnel to work in a selected manner with scheduled maintenance activities, even over time, and allows the historical heritage of the road system to be enhanced for all categories of traffic.

## 2. Methodology

Figure 1 summarizes the methodology adopted in this research work and applied to a case study of a stone pavement network.

In the following subsections, the main phases of the proposed methodology are discussed.

### 2.1. Hierarchization of the Managed Road Network

The preliminary phase in PMS development is the identification of the network’s hierarchical level, geometry, cross-section, and traffic [54]. As the hierarchical level increases, the cross-sectional area is larger, traffic is more intense, and geometric standards are higher. Urban road networks are divided into primary, secondary, and local networks [55]. Figure 2a shows an example of a hierarchized network in which it is possible to note the reference network, Figure 2b focuses on the reference network; some roads (colored gray) must be excluded from the analysis (minor roads, cul de sac, etc.).

Firstly, the pavement network should be divided into branches (such as streets, parking areas, etc.) and each branch should be divided into sections with consistent characteristics in terms of area or length, structural composition, construction history, traffic, and pavement condition. A sample unit (SU) is any identifiable surface of the pavement section covering an area, as suggested by the ASTM D6433, of 225 +/− 90 sqm. The sample unit is the smallest component of the pavement network [56]. All sample units of a section should be considered to estimate the average PCI of the section; however, this approach is considered expensive and time consuming by road managers. In this regard, the minimum number of sample units that must be surveyed into a section to obtain a reliable PCI estimation (95% confidence) was chosen according to the ASTM D6433 guidelines.

### 2.2. Identification of a Reference Road Network

One of the main objectives of the novel methodology is to manage the whole road network using accelerometers mounted on vehicles (e.g., car or bicycle) in order to evaluate the perceived comfort of the users using the *a_wz_* index according to the ISO2631 standard.

The identified reference network is also investigated using profilometers and visual inspections to determine well-known indices such as IRI and PCI: statistical correlations between *a_wz_* -IRI and *a_wz_* -PCI allow the PPI values on the entire network to be estimated. The effort to find an alternative index (*a_wz_*) is due to the fact that traditional methods are unsuitable for urban areas because of the presence of speed limits, the low horizontal curve radii, the numerous intersections, etc. [57]. Instead, accelerometer measurements and *a_wz_* calculation could be more appropriate for urban applications because they can describe perceived comfort conditions in vehicles traveling on the roads [31].

### 2.3. Data Collection and Database Population

Data collection requires a system that is as automated as possible; therefore, a standard-like [39] data collection system is implemented employing QGIS [58] for distress detection; next is useful to install QField [59] software for Android tablets to use during the field test. Figure 3 shows the data entry form for the survey phase; GNSS position is automatically filled in with GNSS sensor data. Information about the sample unit under investigation is also retrieved automatically by cross-referencing positions with information from the GIS graph.

Next, it is possible to switch to the next tab of the tool by selecting the menu named “Distress” (Figure 4). This form allows the distress during the survey to be directly recorded; by selecting the detected distress from the menu, the catalogue’s photos with the description appear automatically. Qualified personnel measure the size of the distress and record the value in the corresponding field.

Then, a photo captured via the mobile device can be uploaded for each identified distress. Finally, it is possible to share the survey data directly within the GIS mapping using QField’s cloud functions. The QGIS software used has the feature to link via plug-in spreadsheets with GIS vectors; therefore, by linking the vector containing the distress to a spreadsheet it was possible to calculate the PCI index of each reference unit. Figure 5 shows the representation of the PCI survey results.

Similarly, to add profilometric measurements and calculate the IRI values, GIS tools allow the process to be simplified and the information overlaid. In particular, in this study, IRI values were calculated using a class I profilometer [60]. A Bluetooth GNSS sensor was placed on the top of the profilometer, which saves the location every three seconds and produces an output file directly imported into GIS. In this way, we obtained the representation of the trajectory followed during the field tests (represented by the arrows in Figure 6).

The measured longitudinal profiles are then processed to return the corresponding IRI values. Through the form in Figure 7, the results of the survey and subsequent processing are stored; specifically, the IRI values, the date, the profile detected by the profilometer, and an image of the pavement during the survey are stored.

### 2.4. Traditional Pavement Evaluation Methods Adopted in This Study

The PCI procedure adopted in this study for stone pavements, which is very similar to ASTM D6433 [39], was proposed by Zoccali et al. [26]. In this research, the PCI calculation procedure was implemented on a Microsoft^®^ Excel spreadsheet based on distress data collected on the road using an Android tablet, as seen in Section 2.3.

A different monitoring technique to evaluate road pavement surface conditions is based on the longitudinal profile analysis in order to calculate IRI values. The computation of IRI is based on a mathematical model known as the quarter car and its mechanical parameters are defined in the ASTM 1926 [61]. In particular, IRI is calculated according to Equation (1):(1)IRI=1L∫0LVz˙s−z˙udt
where *L* is the profile length in km, *V* is the simulated speed equal to 80 km/h, ${\dot{z}}_{s}$ is the time derivative of vertical displacement of the sprung mass, and ${\dot{z}}_{u}$ is the time derivative of the vertical displacement of the unsprung mass. The output of this calculation is an index that increases when road roughness increases, and the commonly recommended units are meters per kilometer (m/km) or millimeters per meter (mm/m).

Differently from the PCI, the IRI standard does not provide ratings that relate the index values to the maintenance interventions; however, common acceptance values are available in the literature for flexible and rigid pavements [20,31,62,63]. On the other hand, regarding stone pavements, IRI threshold values have been derived from the study by Zoccali et al. [26].

### 2.5. Inertial-Based System Adopted in This Study

In addition to the traditional methods (i.e., PCI and IRI), alternative methods based on the measurement of the acceleration profiles have been proposed over time [32,33,34,35,64].

In this study, acceleration data have been collected using the two identical prototypes validated by a recent study [57]. In particular, the device used is a Raspberry Pi [65] microcomputer that interfaces with an IMU unit and a GNSS module (Figure 8). The adopted IMU unit is the InvenSense MPU-9250 [66], and it consists of an accelerometer and a gyroscope that measure both linear acceleration and angular velocity. The GNSS sensor used is a NEO-6M [67], which receives the C/A signal on the L1 carrier from the GPS constellation.

The frequency of data collection is different in both components: in the case of the GNSS sensor, there is a frequency of 1 Hz; for the IMU unit, the maximum mean sample rate effectively obtained was about 83 Hz owing to hardware and software limitations [57].

#### Whole Body Vibration-ISO 2631

The frequency weighted acceleration (*a_wz_*) depends on the vertical acceleration measured inside a vehicle in motion to assess the ride comfort of road users and not directly on the defects of the pavement.

The acceleration time-history collected in the field tests by the adopted prototype was processed using a code written in Matlab^®^ language in order to calculate the *a_wz_* index values every second, whereby the signal is divided each 2 s, and 1 s of overlap is considered. Regarding the evaluation of the index, starting from the accelerations in the time domain, it is possible to determine the frequency range’s relative spectrum and calculate the root mean square deviation of the spectral accelerations (RMS). The ISO 2631-1 suggests the frequencies between 0.5 and 80 Hz as the cause of comfort decay. This range is divided into a one-third octave spectrum (*a_w1_*, *a_w2_*, …, *a_w23_*). The *a_wz_* value is calculated according to Equation (2):(2)$$a_{wz}=\sqrt{\sum_{i=1}^{23}{\left(W_{k,i}\cdot a_{z,i}\right)}^{2}}$$
where *W_k,i_* is the *i*-th frequency weighting in one-third octave bands for the sensor, provided by the ISO2631 standard, and *a_z,i_* is the vertical RMS acceleration for the *i*-th one-third octave band. The ISO 2631 also suggests the threshold values for public transport as reported in Table 1.

It should be noted that the sample unit lengths generally vary from 20 to 80 m, considering an average travel speed of about 5–6 m/s, within each SU fall between 3 and 16 *a_wz_* measurements for each survey: for each SU, the *a_wz_* value is returned by the average of the values. GIS tools manage spatial analysis algorithms, making it possible to associate points with sample units (areas). Therefore, it is possible to switch from the point representation of the *a_wz_* values (Figure 9a) to the sample unit representation (Figure 9b). The rating scale adopted is provided by ISO 2631 [30].

In this study, *a_wz_* values have been calculated in the whole stone road network. However, they do not characterize the maintenance level of the network making it necessary to find regression models for the adopted PPIs.

### 2.6. Assessment of Pavement Condition with the Estimated PPI Indices

The values of PCI, IRI, and *a_wz_* are available for the sample units of the reference network: correlations among these indices have been investigated. Previous studies [31,68,69] have shown the existence and usefulness of these regressions for flexible pavements: in this study, we would also like to identify similar regressions for stone pavements (Sampietrini). In particular, an exponential-type regression model: the choice of this model is supported by a previous study conducted on stone pavements [26]. The studies [31,67] show that regressions should be derived considering a constant speed value; in this paper, it was decided to find regressions at a speed of 20 km/h. In order to use one tool for all phases, regression analysis was also conducted in GIS.

## 3. Field Tests

The field test was carried out in the city of Velletri, located in the southern district of Rome. The historic center is paved with Sampietrini pavements, consisting of polished black basalt blocks placed side by side in a regular pattern (Figure 10). The typical shape is cubic or a square-based truncated pyramid solid measuring 12 × 12 cm and 18 cm high.

The urban road network, consisting of about 12 km of roads with irregular cross-sections, is extremely non-homogeneous in terms of maintenance, traffic, and construction methodology. The hierarchy identification was mostly based on traffic surveys, assigning a higher hierarchy as the traffic intensity increases.

For the graph construction, we started with data from the 2014 regional technical map of the Italian Lazio region [70] provided in shape file format. However, the regional map identifies the streets without distinguishing between the different types of pavements, making it necessary to detect which streets in the city were paved with Sampietrini. This problem was overcome by using Street View^TM^ [71] and some field visual inspections. GIS representation of Velletri’s Sampietrini network is shown in Figure 11.

Firstly, the stone-paved network areas were identified. The next step was to divide the roads according to the criteria outlined in the ASTM D6433. The identification of the sample units was carried out using the free software QGIS, assigning their toponymy and a progressive number. A total of 235 sample units were considered in Velletri’s Sampietrini network, and they were identified uniquely, as shown in Figure 12.

Then, the proposed methodology required the identification of a reference network; in this case study, 10 representative sample units were identified by choosing different levels of pavement deterioration.

The vertical accelerations recorded both in a car with a mass of about 1050 kg and with a mountain bike (suspension-less) were processed in order to calculate *a_wz_* index values. In particular, an effort was made to maintain a speed of 20 km/h for both vehicles.

The position of Raspberry-based sensors needs a fixed location. In addition, it should provide good satellite coverage and should represent as closely as possible the effective comfort conditions perceived by the user; the preferred options resulted as:
The sensors were externally on the vehicle’s frame; this required the construction of a magnetic case that allows the prototype to be placed on the car body, as shown in Figure 13;As for the bicycle, the measurement points were chosen on the bicycle’s frame (Figure 14a) using the same socket used for the car, and on the cyclist’s helmet (Figure 14b) in a vertical position relative to the rider’s head and without a socket.

After the survey phase of the reference network was concluded, we characterized the remaining network only through the *a_wz_* index. In this regard, routes with fewer turning maneuvers were chosen at a preliminary stage and represented in GIS mapping (Figure 15).

An important point to mention is that although the adopted inertial-based system is equipped with a GNSS receiver, in urban areas, the positioning errors increase due to the phenomenon of urban canyons [72,73,74]. For this reason, the development of a corrective algorithm in GIS environments to relocate the points acquired by the sensors to the centerline of the road was required.

## 4. Results and Discussion

In this study, once the field tests are concluded, the *a_wz_* index is known on the entire inspectable network (closed roads, cul de sacs, and narrow roads are excluded). On the other hand, for the 10 sample units of the reference network, the *a_wz_*, IRI, and PCI indices are computed.

### 4.1. The Index a_wz_

The *a_wz_* index surveys were conducted by car and bicycle, Figure 16 shows the distribution of the *a_wz_* values referring to the field tests conducted with the car.

Figure 16a shows the *a_wz_* results in terms of colored points, in accordance with Section 2.5. Figure 16b represents the next step, which is from the point representation to the sample unit representation.

It can be observed in Figure 16b that about 70% of the sample units are included in the range from “a little uncomfortable” to “fairly uncomfortable” of the ISO 2631 standard (as already defined in Table 1).

A similar procedure was carried out for the bicycle reliefs, and the results are shown in Figure 17.

Comparing Figure 16b with Figure 17, the *a_wz_* values are significantly higher than those calculated in the car: this is due to the differences in damping between the two vehicles. In addition, comparing Figure 17a and Figure 17b, we observe that the sensor placed on the frame measures significantly higher acceleration values than the one collected by the helmet sensor; this can be explained by the additional damping offered by the seat and the human body. It should also be noted that bicycle measurement values are significantly higher than the ranges considered comfortable by the reference standard [30], suggesting that the thresholds provided for public transportation cannot be applied to bicycles. As a confirmation of this, the study [75] conducted statistical investigations to associate the levels of the *a_wz_* index with the level of comfort perceived by bike riders, showing that it is possible to accept values that are significantly higher than those provided by the reference standard.

### 4.2. Statistical Regression between Indices (a_wz_-PCI, a_wz_-IRI) and Error Estimation

Once the measurement campaign was concluded, the PCI, IRI, and *a_wz_* values for the reference network were obtained, while for the remaining network only the values of *a_wz_* were known. Then the regression functions between *a_wz_* and PCI indices (Figure 18) and *a_wz_* and IRI indices (Figure 19) were derived.

As for the bicycle measurements, three reference sample units were excluded because a constant speed of 20 km/h could not be maintained, so the corresponding *a_wz_* values obtained were not considered.

It is possible to notice that in relation to the measurement campaigns with the bicycle, statistically weaker regressions were derived, and this occurred due to the difficulty in keeping a uniform speed. To quantify this dispersion in the reference network, additional statistical analysis was conducted; in addition to the mean value of the reference sample unit, the standard deviation was also computed for the *a_wz_* index. Therefore, the coefficient of variation “CV”(Figure 20) was used as a statistical indicator of confidence.

The coefficients of variation calculated from the car surveys are significantly lower than from the bicycle surveys; with the exception of sample units #7 and #10, which, having traveled downhill, surveyed the bicycle at a constant speed of 20 km/h, thus obtaining more reliable results.

Further analysis took into account the obtained values of R^2^ (Figure 18 and Figure 19) with reference to the number of sections considered for evaluation, through standard statistical tests (ANOVA) of the goodness of the statistical models. In particular, for the case study, the statistical tests were reported for the vehicle and for the bicycle, distinguishing between the performance for predicting the PCI-*a_wz_* (Table 2) and IRI-*a_wz_* (Table 3) relationships.

It is well-known that Significance F is used to evaluate if the regression model is statistically significant [76]. In this analysis, a value of 0.01 (1%) was established as the significance level; therefore, to test if the results shown are statistically relevant Significance F should be less than 0.01; only for the car this condition is satisfied and for both PCI-*a_wz_* and IRI-*a_wz_*_,_ highlighted with red text in the tables. This result can be seen as a consequence of the better speed control obtained during the surveys with the car.

### 4.3. PCI and IRI Values of the Entire Network

Due to the lower confidence of the regressions obtained from the bicycle surveys, it was decided to use only the car surveys for the characterization of the entire network. The regression functions in Figure 18 and Figure 19 were implemented directly in GIS, thus obtaining a vector containing the estimated PCI and IRI and a representation by performance classes (Figure 21).

With regard to the threshold values, the following were considered:
The guidelines provided in the ASTM D6433 for the PCI.Regarding the IRI index, there are no commonly accepted scale ratings, so the thresholds reported in a previous study by Zoccali et al. [26], related to the speed of 20 km/h, were used in this study.

## 5. Conclusions

The management of historic stone-paved roads is still an open challenge.

This article proposes a novel method for the quick monitoring of the road surface of stone pavements using accelerometer sensors mounted on the car and bicycle in order to calculate the *a_wz_* values. PCI and IRI pavement evaluation methods were also performed in a reference network in order to provide regression models that can be used to obtain indications about the state of health of the examined network

In addition, GIS was provided as a tool to support data collection and management in an easy way. The implementation of the road network in GIS enabled the surveys to be represented intuitively in a graphical database.

The novel methodology was applied to a case study in an Italian historic town and the results proved that it can be used to assess stone pavement conditions on the whole urban road network. The proposed method allows the assessment of the stone pavement road network both through direct accelerometric measurements with the *a_wz_* parameter and through statistical models that estimate the conditions of the pavement with traditional performance indicators (IRI and PCI).

Correlations between the PPIs were analyzed in the reference network, obtaining results comparable with the study carried out by Zoccali et al. [26] In particular, for the same investigated sections, measurements collected in both cars and bicycles showed that, with the same level of maintenance, as the damping offered by the vehicle decreases, the value of the *a_wz_* index increases. However, there is a greater propensity for users to tolerate higher stresses.

It should then be noted that the correlations found from the data of the car surveys were more reliable (R^2^_PCI-awz_ = 0.96, R^2^_IRI-*awz*_ = 0.74) than the correlations found for the surveys carried out on the bicycle both on the helmet (R^2^_PCI-*awz*_ = 0.32, R^2^_IRI-*awz*_ = 0.44) and on the frame (R^2^_PCI- *awz*_ = 0.37, R^2^_IRI- *awz*_ = 0.55). In addition, in-depth statistical analyses were conducted with the aim of checking the reliability of the predictions obtained; in particular, the Significance F values showed that the regression model proposed in the case of the car is statistically relevant (Significance F < significance level set to 0.01), while the models for the bicycle—where speed control was proven to be a problem—showed less accuracy.

Therefore, the results proved that the proposed methodology, based on GIS technology, can be used to assess the stone pavement conditions of road networks characterized by stone pavements.

Remaining within the scope of methodology refinement, it could be envisaged to filter the results from disturbances such as, e.g., hypothetically lying stones, boards, bottles, or other objects of different geometry on an even surface.

## Figures and Tables

**Figure 1 sensors-22-06560-f001:**
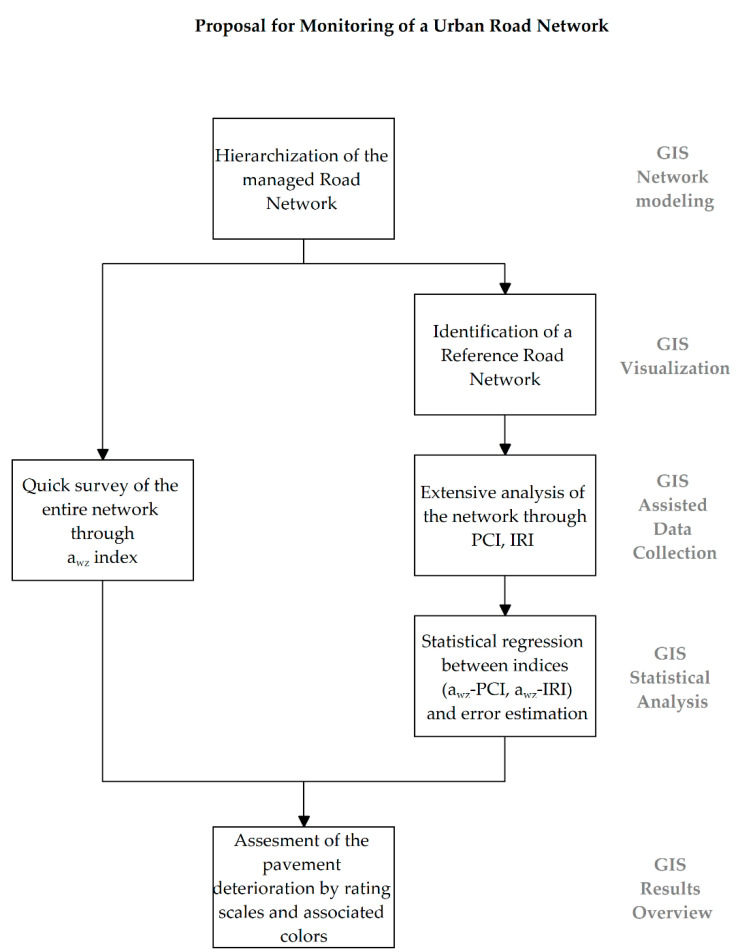
Flowchart of the proposed methodology.

**Figure 2 sensors-22-06560-f002:**
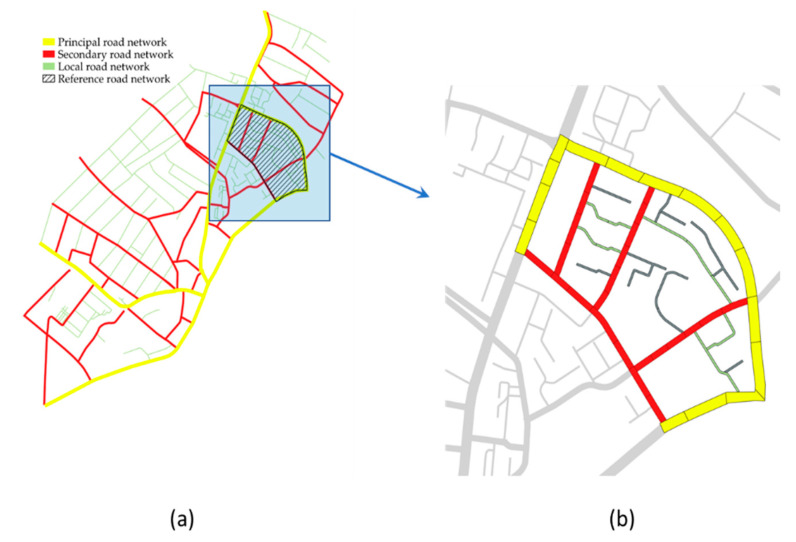
Road Network: (**a**) entire network (**b**) reference network.

**Figure 3 sensors-22-06560-f003:**
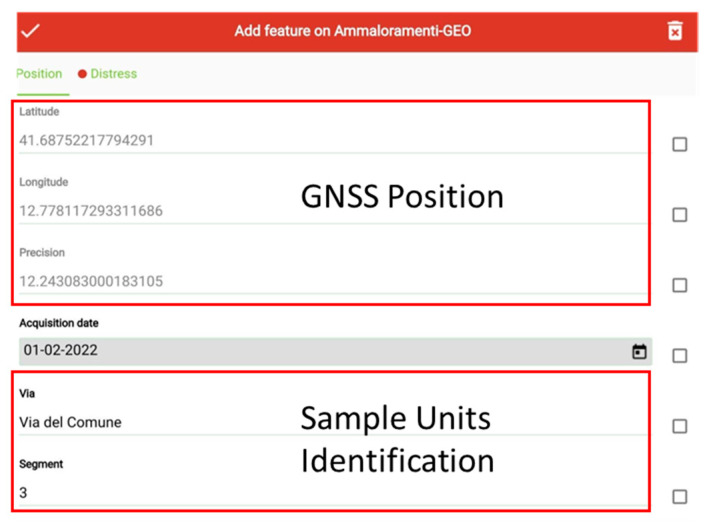
Position form.

**Figure 4 sensors-22-06560-f004:**
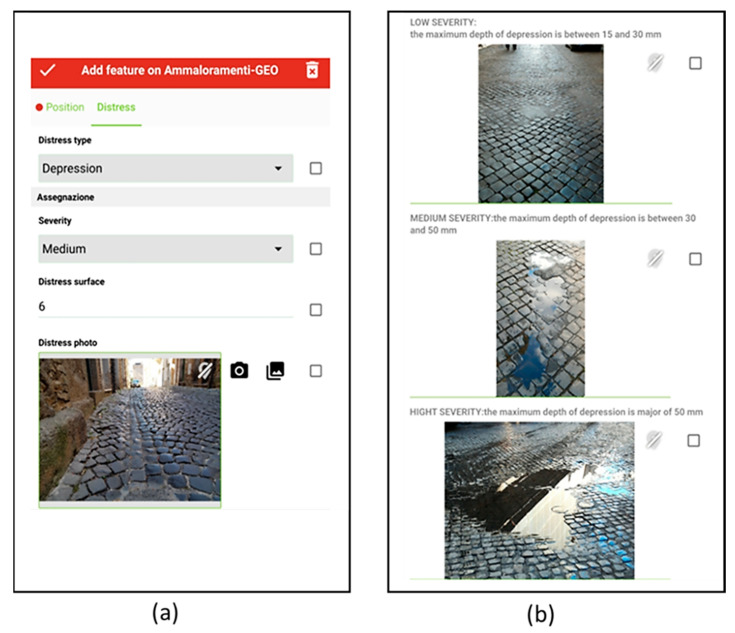
Distress identification menu: (**a**) data input with the possibility to add in situ the photo of the defect; (**b**) photos of the different defects specialized for stone pavements [26], to support the operator in identifying the observed defect.

**Figure 5 sensors-22-06560-f005:**
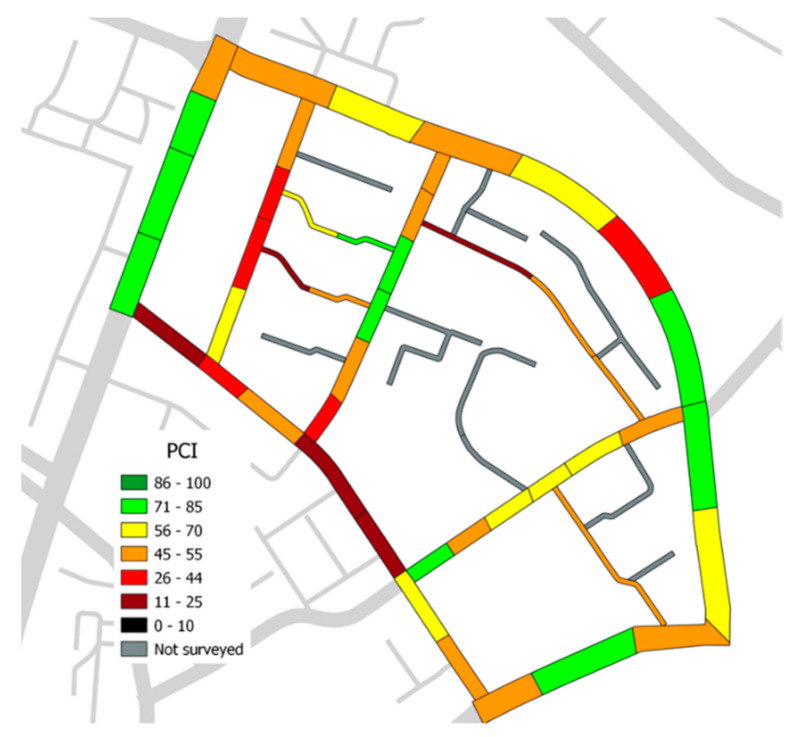
Visualization of PCI results in GIS.

**Figure 6 sensors-22-06560-f006:**
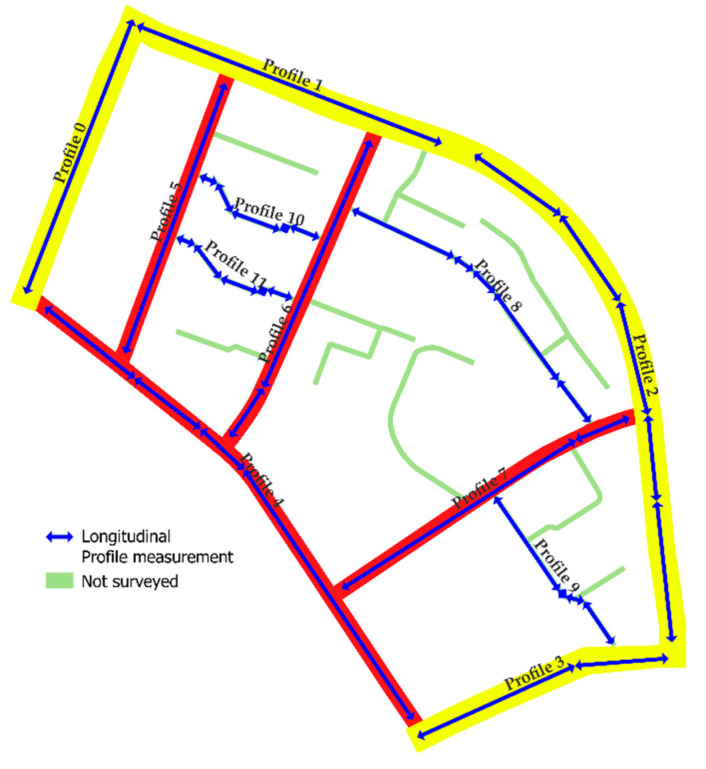
Visualization of IRI results in GIS.

**Figure 7 sensors-22-06560-f007:**
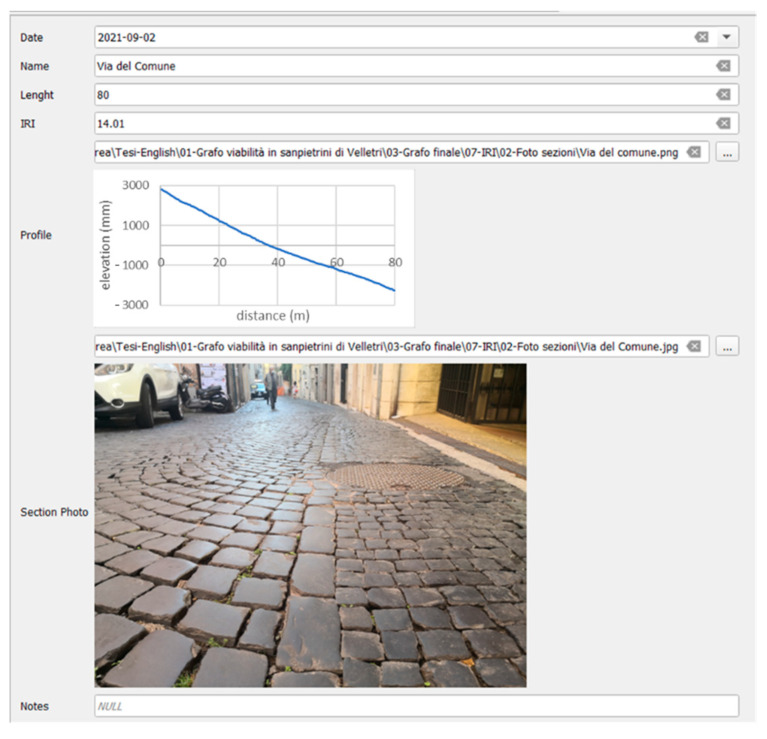
IRI form.

**Figure 8 sensors-22-06560-f008:**
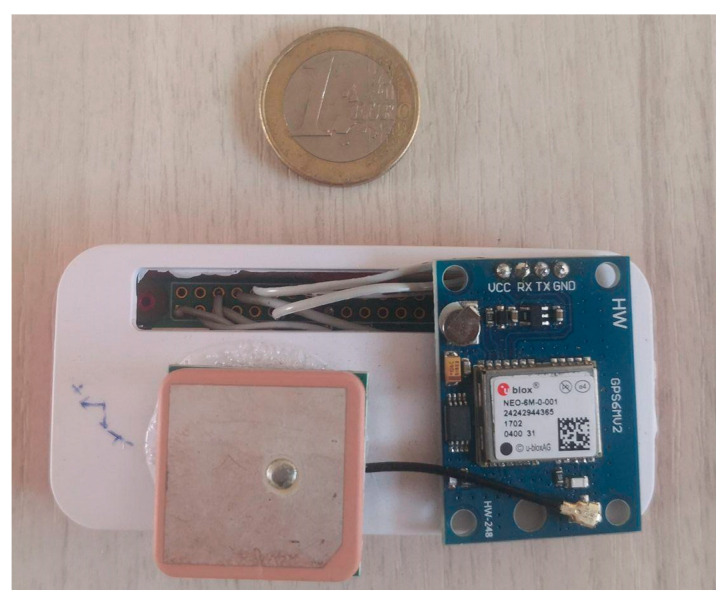
Raspberry-based IMU device.

**Figure 9 sensors-22-06560-f009:**
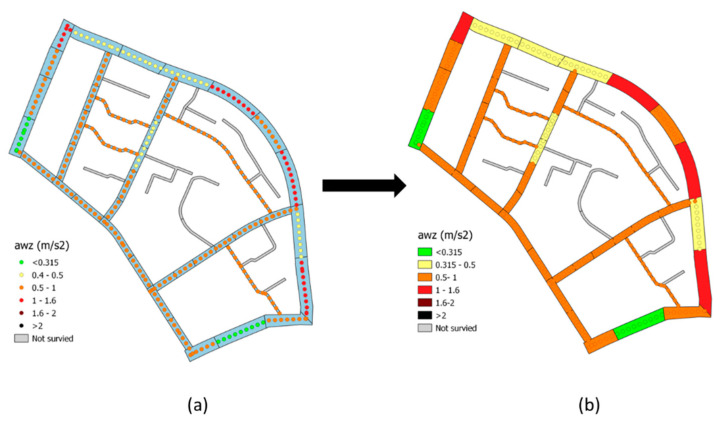
Visualization of *a_wz_* results in GIS: (**a**) point representation (**b**) sample unit representation.

**Figure 10 sensors-22-06560-f010:**
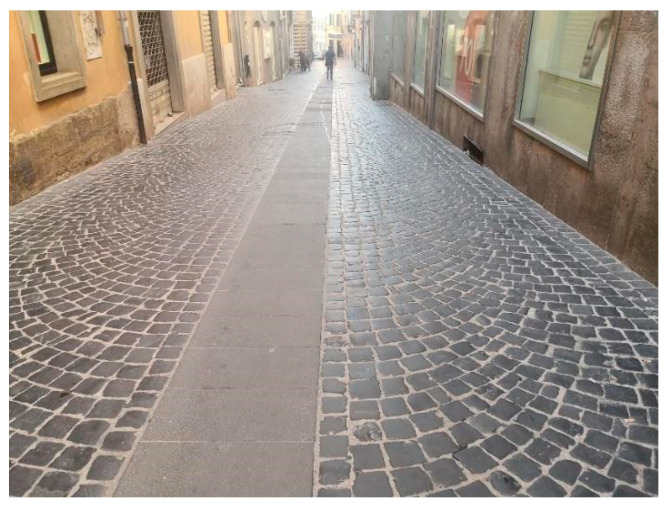
Typical Sampietrini Pavement in the city of Velletri.

**Figure 11 sensors-22-06560-f011:**
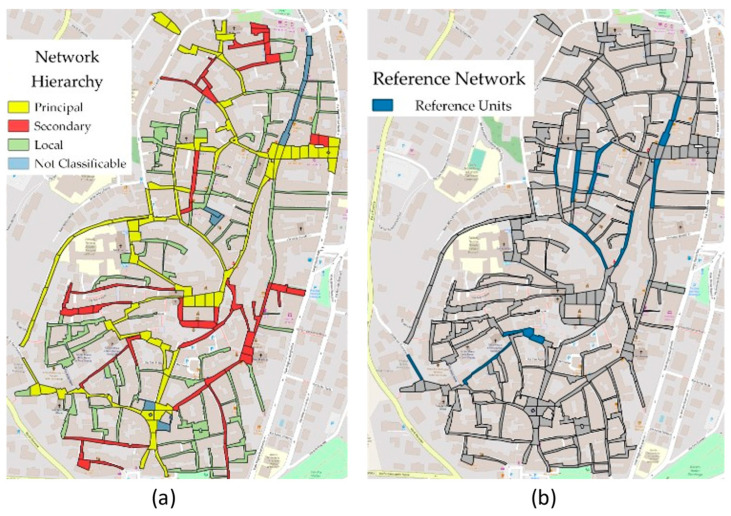
Stone pavement network in Velletri: (**a**) network hierarchy (**b**) reference network.

**Figure 12 sensors-22-06560-f012:**
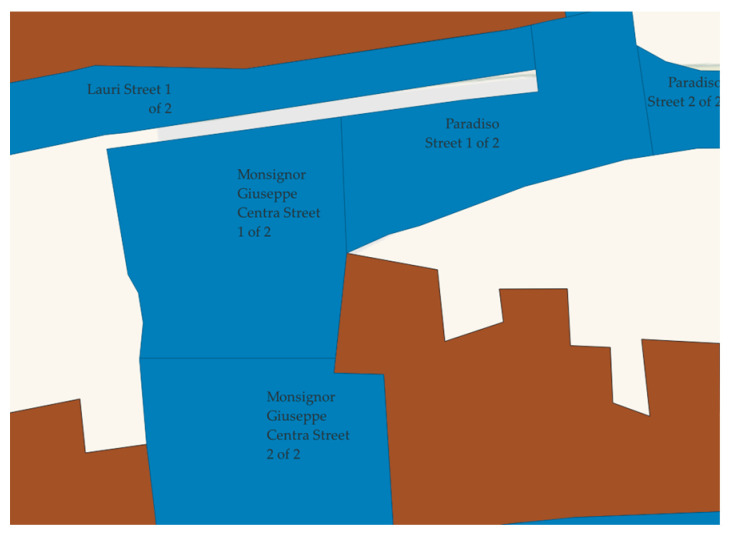
GIS visualization of some sample units of the Sampietrini network in Velletri.

**Figure 13 sensors-22-06560-f013:**
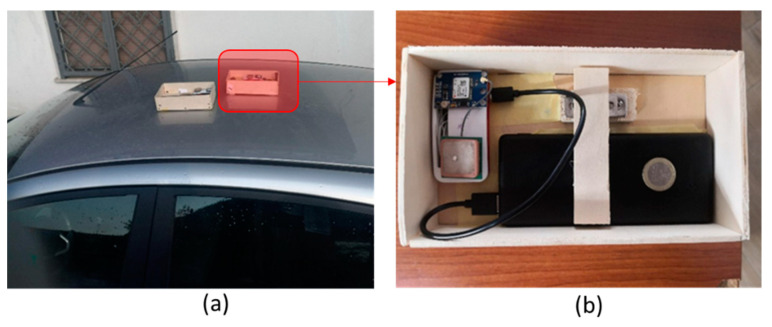
Installation of the prototypes on the car body: (**a**) position of the sensor and (**b**) magnetic case.

**Figure 14 sensors-22-06560-f014:**
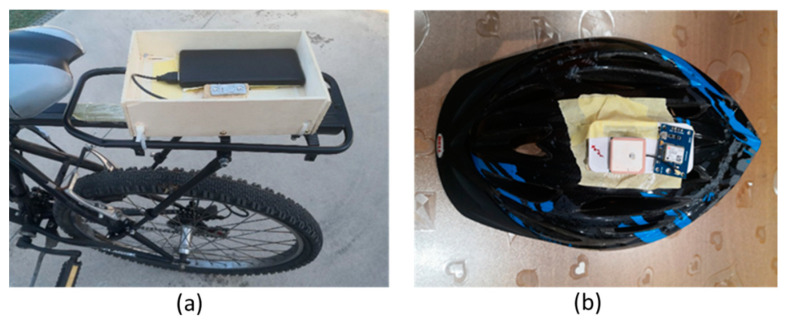
Installation of the prototypes on the bike: (**a**) Bike-frame and (**b**) Bike-helmet details.

**Figure 15 sensors-22-06560-f015:**
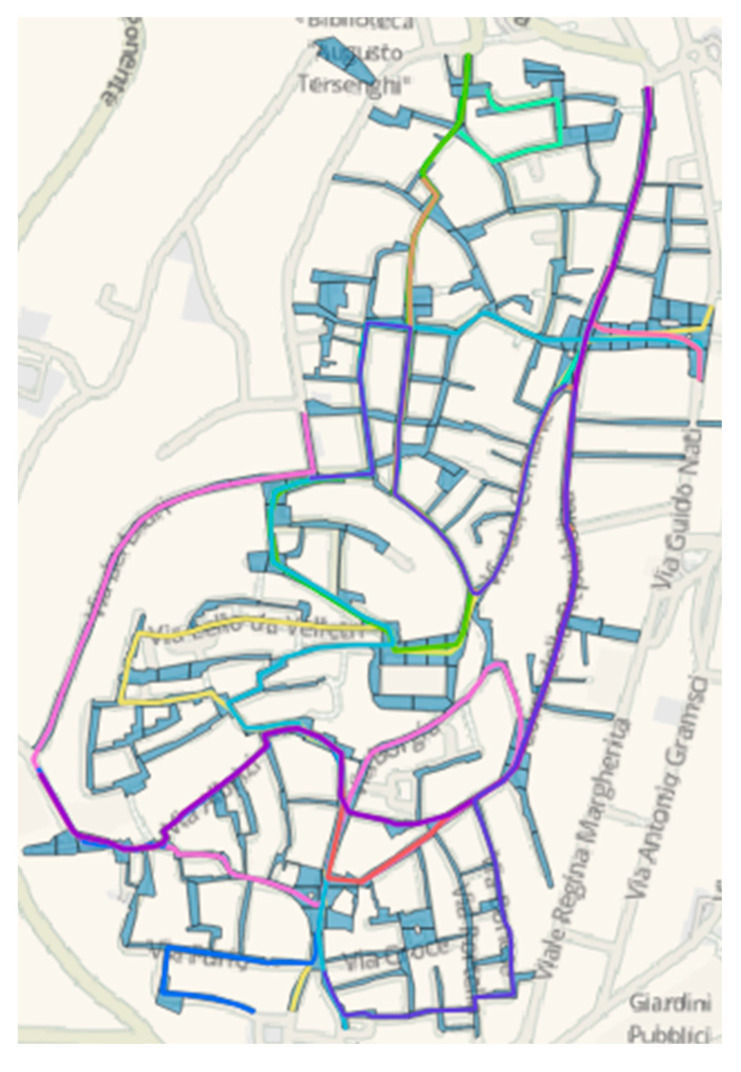
Inspection routes.

**Figure 16 sensors-22-06560-f016:**
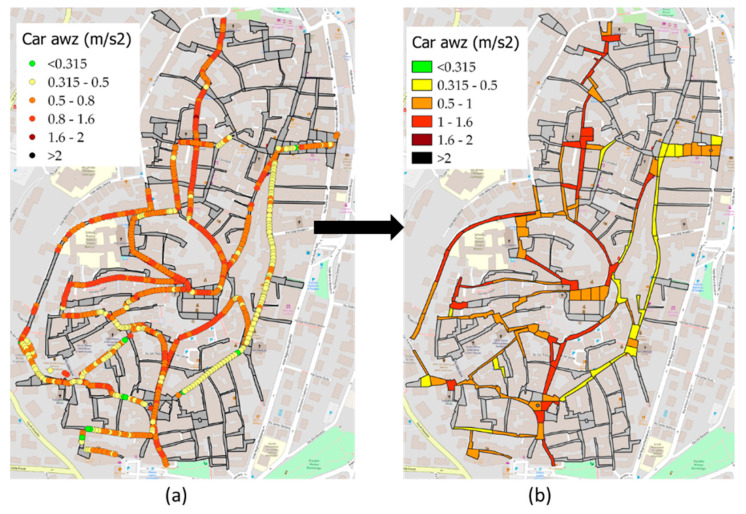
*a_wz_* car results in terms of: (**a**) points and (**b**) areas.

**Figure 17 sensors-22-06560-f017:**
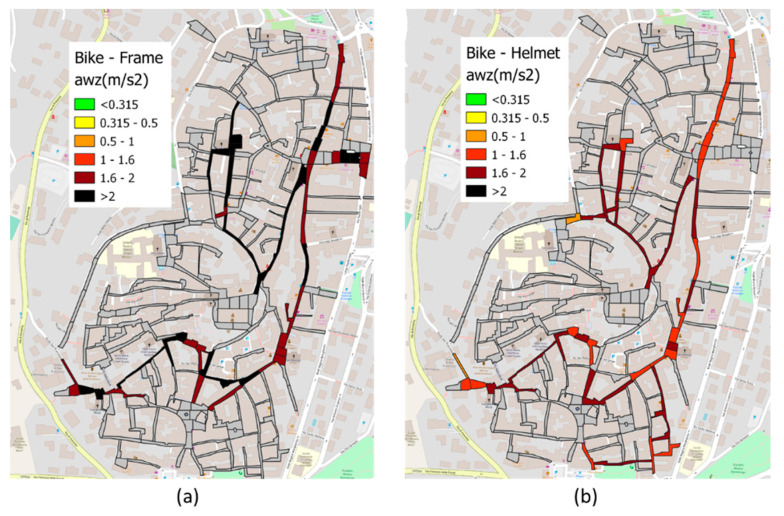
*a_wz_* bike results: (**a**) bike frame results (**b**) bike helmet results.

**Figure 18 sensors-22-06560-f018:**
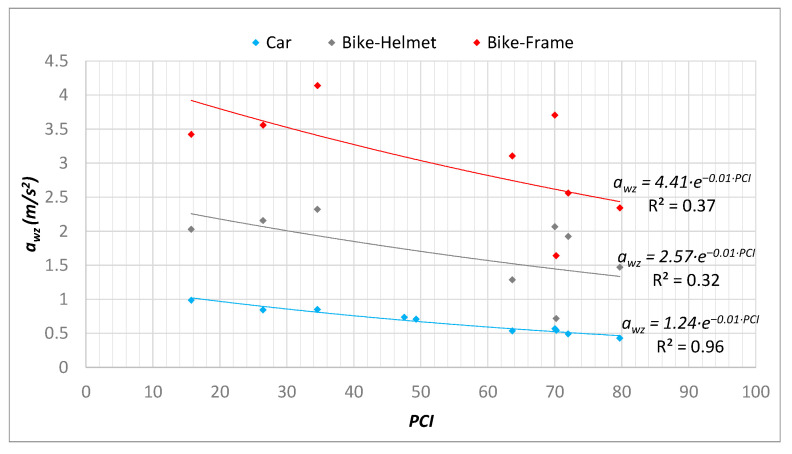
*a_wz_*-PCI regression considering different vehicles and sensors positions.

**Figure 19 sensors-22-06560-f019:**
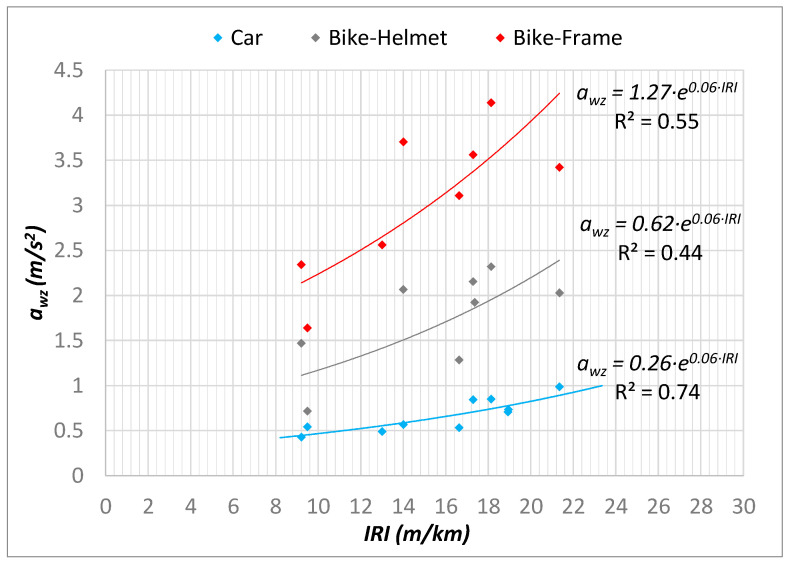
*a_wz_*-IRI regression considering different vehicles and sensors positions.

**Figure 20 sensors-22-06560-f020:**
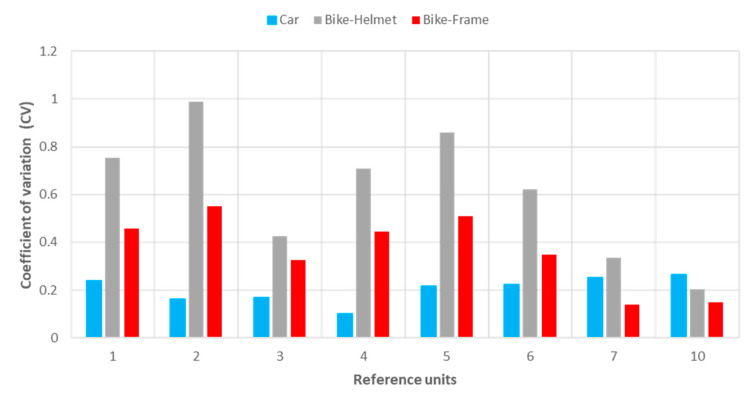
Coefficients of variations related to the *a_wz_* values calculated in the sample units surveyed in bicycle and car during the field tests.

**Figure 21 sensors-22-06560-f021:**
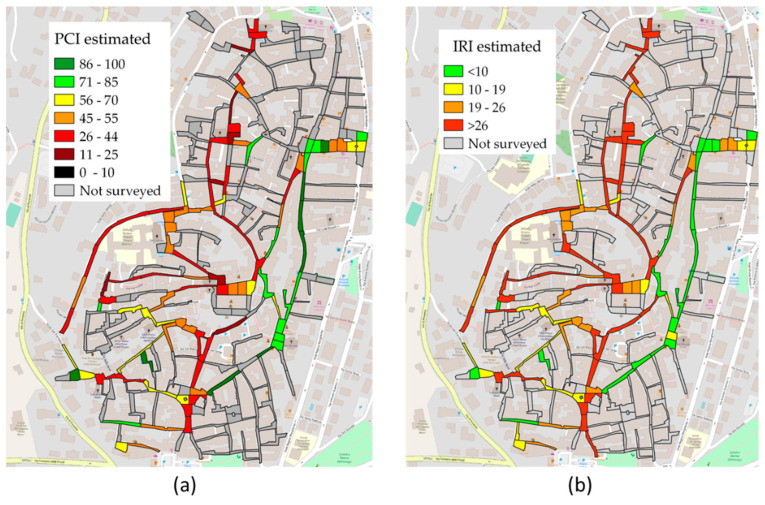
(**a**) PCI and (**b**) IRI estimated for the surveyed network using regression models.

**Table 1 sensors-22-06560-t001:** Comfort levels and *a_wz_* threshold values according to ISO 2631-1 standard.

*a_wz_* Values	Comfort Level
Less than 0.315 m/s^2^	Not uncomfortable
0.315–0.63 m/s^2^	A little uncomfortable
0.5–1.0 m/s^2^	Fairly uncomfortable
0.8–1.6 m/s^2^	Uncomfortable
1.25–2.5 m/s^2^	Very uncomfortable
Greater than 2.0 m/s^2^	Extremely uncomfortable

**Table 2 sensors-22-06560-t002:** Statistical parameters used in evaluating the performance for predicting the PCI-*a_wz_* relationship.

		Sum of Squares (SS)	Mean Square (MS)	MS Regres./ MS Resid. (F)	Significance F
Car	Regression	0.295	0.295	286.0	1.52 × 10^–7^
	Residual	0.008	0.001		
	Total	0.303			
Bike-Helmet	Regression	0.673	0.673	2.9	1.39 × 10^–1^
	Residual	1.386	0.231		
	Total	2.059			
Bike-Frame	Regression	1.856	1.856	3.9	9.73 × 10^–2^
	Residual	2.890	0.482		
	Total	4.745			

**Table 3 sensors-22-06560-t003:** Statistical parameters used in evaluating the performance for predicting the IRI-*a_wz_* relationship.

		Sum of Squares (SS)	Mean Square (MS)	MS Regres./ MS Resid. (F)	Significance F
Car	Regression	0.213	0.213	19.0	2.41 × 10^–3^
	Residual	0.090	0.011		
	Total	0.303			
Bike-Helmet	Regression	0.980	0.980	5.4	5.83 × 10^–2^
	Residual	1.079	0.180		
	Total	2.059			
Bike-Frame	Regression	2.922	2.922	9.6	2.11 × 10^–2^
	Residual	1.824	0.304		
	Total	4.745			

## Data Availability

The data presented in this study are available on request from the corresponding author. The data are not publicly available due to confidentiality reasons.

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
