# Peer review of "Development of a GIS-Based Methodology for the Management of Stone Pavements Using Low-Cost Sensors"

_sensors, 2022, doi:10.3390/s22176560_

Round 1
Reviewer 1 Report
Dear Authors
The paper: entitled Development of a GIS-Based Methodology for the Management of Stone Pavements using low-cost sensors is very interesting and has a utilitarian meaning, bringing specific social benefits. It also serves to preserve the cultural heritage through identification and preventive actions in the field of maintenance of stone road surfaces. The authors of the work took up an unusual topic, but very interesting and important all over the world. My notes for publication are:
- the authors have made an extensive review of the literature in the field of the discussed subject, but its content raises many reservations, please follow the guidelines with the notation of references (very long links to internet references, e.g. ref. 63-65, 68 etc. please minimize the link using another website or refer to materials published as articles.)
- in Figure 1, the arrows at the bottom of the diagram should be corrected
- usually when riding a bicycle or a car, we avoid surface unevenness, how was the problem of objective identification of these inequalities solved if we avoid them?
- what is the measurement uncertainty of the measuring systems used with low-cost acceleration sensors
- in Figure 4, add indexes of photos a) ... b) .... c) ... etc. --- see fig. 14
- How do the authors separate vibrations from the work of shock absorbers in the car (or bicycle) from uneven surfaces, how is the signal filtered and is it done online?
- in Figure 13, add indexes a) ... b) --- see Fig. 14
- it is worth thinking about a different position of the sensors than on the roof or on the trunk or the helmet, maybe a better place for the wheels that have direct contact behind the surface or the use of a differential measurement method
- the caption under Figures 18 and 19 should be extended
- the regression equations also need to be corrected so that the right side of the equation is equal to the left side in terms of units, use the appropriate variables (y = awz (PCI) so adjust your regression equation ........, y = awz (IRI) therefore adjust your regression equation), add unit factors to order the units because m / s2 cannot equal m / km
- adjust the graph from Figure 20 to the remaining ones, e.g. fig. 18 or fig. 19
- what influence does the speed of movement of the vehicle with the sensor have on the result of the identification of unevenness of the road surface, or does it have any influence at all? Therefore, the uncertainty of measurement should be evaluated.
- how the measurement results are filtered from disturbances such as e.g. hypothetically lying stones, boards, bottles or other objects of different geometry on an even surface
In the conclusions, the authors should refer both qualitatively and quantitatively to the obtained results in the context of the assessment of measurement uncertainty. They should demonstrate the quantitative results and the likelihood of certainty of these results. I missed the advantages and disadvantages of the presented method.
Kind regards
Reviewer
Author Response
Manuscript ID: sensors-1878271
Development of a GIS-Based Methodology for the Management of Stone Pavements Using Low-Cost Sensors
Answer to Reviewers’ Comments
First of all, we thank the Reviewers for their valuable comments and suggestions that helped us to improve the quality of our paper.
We revised the manuscript taking into account all the reviewers’ concerns.
Reviewer #1:
Reviewer #1:
Dear Authors
The paper: entitled Development of a GIS-Based Methodology for the Management of Stone Pavements using low-cost sensors is very interesting and has a utilitarian meaning, bringing specific social benefits. It also serves to preserve the cultural heritage through identification and preventive actions in the field of maintenance of stone road surfaces. The authors of the work took up an unusual topic, but very interesting and important all over the world. My notes for publication are:
- the authors have made an extensive review of the literature in the field of the discussed subject, but its content raises many reservations, please follow the guidelines with the notation of references (very long links to internet references, e.g. ref. 63-65,68 etc. please minimize the link using another website or refer to materials published as articles.).
Answer 1.1: We thank the reviewer for the valuable suggestion. We modified the notation of references minimizing the links.
- in Figure 1, the arrows at the bottom of the diagram should be corrected.
Answer 1.2: The Reviewer is right. The authors modified Figure 1.
- usually when riding a bicycle or a car, we avoid surface unevenness, how was the problem of objective identification of these inequalities solved if we avoid them?
Answer 1.3: We thank the reviewer for the useful comments, which allows us to clarify this important point. In this regard, we have added the following sentence at lines 81-86:
“The road surface detected is determined by the number of surveys, due to the well-known vehicles’ trajectory dispersion phenomena [52,53]: the proposed approach makes it easy to carry out multiple measurements for the investigated area, as the system is easy to install on several vehicles and does not require qualified personnel, and thus detect the entire lane.”
- what is the measurement uncertainty of the measuring systems used with low-cost acceleration sensors
Answer 1.4: In this paper it has been adopted a low-cost pavement monitoring inertial-based system validated in a previous study [Loprencipe, G.; de Almeida Filho, F.G.V.; de Oliveira, R.H.; Bruno, S. Validation of a Low-Cost Pavement Monitoring Inertial-Based System for Urban Road Networks. Sensors 2021, 21, 3127, doi:10.3390/s21093127] employing concomitant measurements made with the proposed sensor and a precision measuring instrument.
- in Figure 4, add indexes of photos a) ... b) .... c) ... etc. --- see fig. 14
Answer 1.5: We thank the reviewer for the useful comments. The authors modified Figure 4 and also the caption.
- How do the authors separate vibrations from the work of shock absorbers in the car (or bicycle) from uneven surfaces, how is the signal filtered and is it done online?
Answer 1.6: We thank the reviewer for the valuable suggestion, which allows us to clarify this important point. In this regard, we have added the following sentence at lines 221-223:
“The frequency weighted acceleration (awz) depends on the vertical accelerations measured inside a vehicle in motion to assess the ride comfort of road users and not directly the defects of the pavement.”
- in Figure 13, add indexes a) ... b) --- see Fig. 14
Answer 1.7: We thank the reviewer for the useful comments. The authors modified Figure 13.
- it is worth thinking about a different position of the sensors than on the roof or on the trunk or the helmet, maybe a better place for the wheels that have direct contact behind the surface or the use of a differential measurement method
Answer 1.8: We thank the reviewer for the useful comments. We have added the text at lines 295-303:
“ In addition, it should provide good satellite coverage and should represent as closely as possible the effective comfort conditions perceived by the user; the preferred options resulted to be:
- The sensors were externally on the vehicle’s frame; this required the construction of a magnetic case that allows the prototype to be placed on the car body, as shown in Figure 13;
- As for the bicycle, the measurement points were chosen on the bicycle’s frame (Figure 14a) using the same socket used for the car, and on the cyclist's helmet (Figure 14b) in a vertical position relative to the rider's head and without a socket. ”
- the caption under Figures 18 and 19 should be extended
Answer 1.9: We thank the reviewer for the useful comments. The authors extended the caption under Figures 18 (“awz-PCI regression considering different vehicles and sensors positions.”, line 358) and Figure 19 (“awz-IRI regression considering different vehicles and sensors positions.”, line 360).
- the regression equations also need to be corrected so that the right side of the equation is equal to the left side in terms of units, use the appropriate variables (y = awz (PCI) so adjust you regression equation ........, y = awz (IRI) therefore adjust you regression equation), add unit factors to order the units because m / s2 cannot equal m / km
Answer 1.10: Theoretically, the reviewer is right, but frequently in the research runs a regression analysis when the variables are measured in different units of measurement. The slope is always the change in the response variable (in whatever unit that is measured in) for a unit change in each predictor variable - for whatever unit that is measured in. So there is no problem except that you have to reflect on what you are doing. At any rate, we use the appropriate variables in the equations reported in the graphs.
- adjust the graph from Figure 20 to the remaining ones, e.g. fig.18 or fig. 19
Answer 1.11: We thank the reviewer for the useful comments. The authors modified Figure 20.
- what influence does the speed of movement of the vehicle with the sensor have on the result of the identification of unevenness of the road surface, or does it have any influence at all? Therefore, the uncertainty of measurement should be evaluated.
Answer 1.12: The ride speed and the type of vehicle have a lot of influence in the evaluation of the awz parameter, for this reason, as we have often reported in the article, during the surveys we always used the same vehicle type, paying attention to keep the speed constant during the surveys. For this reason, we were able to obtain better correlations with the car than with the bicycle. The road sections where measurements were carried out with both systems (traditional surveys and with accelerometers on board the vehicles) were used precisely to calibrate the vehicle characteristics.
- how the measurement results are filtered from disturbances such as e.g. hypothetically lying stones, boards, bottles or other objects of different geometry on an even surface
Answer 1.13: We thank the reviewer for the valuable suggestion: this issue is not included in the present study but is scheduled as part of future works. In this regard, the authors added the text at lines 441-443:
“ Remaining within the scope of methodology refinement, it could be envisaged to filter the results from disturbances such as e.g. hypothetically lying stones, boards, bottles or other objects of different geometry on an even surface.”
- In the conclusions, the authors should refer both qualitatively and quantitatively to the obtained results in the context of the assessment of measurement uncertainty. They should demonstrate the quantitative results and the likelihood of certainty of these results. I missed the advantages and disadvantages of the presented method.
Answer 1.14: It is known that R-squared (R2) is a statistical measure that represents the proportion of the variance for a dependent variable that's explained by an independent variable or variables in a regression model. R-squared can be considered a goodness-of-fit measure for regression models.
In order to demonstrate the quantitative results and the likelihood of certainty of the results, the authors added the text at lines 377-394:
“ Further analysis taken into account the obtained values of R2 (Figures 18 and 19) with reference to the number of sections considered allowing for evaluation, through standard statistical tests, of the goodness of the statistical models. For example, for the case study, the statistical tests have been reported for the vehicle and for the bicycle, distinguishing between the performance for predicting the PCI-awz (Table 2) and IRI-awz (Table 3) relationships.
Table 2. Statistical parameters used in evaluating the performance for predicting PCI-awz relationship
Table 3. Statistical parameters used in evaluating the performance for predicting IRI-awz relationship
It is well-known that the Significance F is used to evaluate if the regression model is statistically significant [76]. In this analysis, a value of 0.01 (1%) was established as the significance level; therefore, to test if the results shown are statistically relevant the Significance F should be less than 0.01: only for the car this condition is satisfied and for both PCI-awz and IRI-awz, highlighted with red text in the tables. This result can be seen as a consequence of the better speed control obtained during the surveys with the car.”
In addition, in the section “Conclusions” the authors added the text at lines 420-423:
“ The proposed method allows the assessment of the road network in stone pavement both through direct accelerometric measurements with the awz parameter and through statistical models that estimate the conditions of the pavement with traditional performance indicators (IRI and PCI).”
and at lines 433-437:
“ In addition, in-depth statistical analyses were conducted with the aim of checking the reliability of the predictions obtained: in particular, the Significance F values showed that the regression model proposed in the case of the car is statistically relevant (Significance F< significance level set to 0.01), while the models for the bicycle - where speed control was proven to be a problem - showed less accuracy.”

Reviewer 2 Report
Fig. 1 - not all arrows are fully visible.
Line 48-50 the language is not quite clear.
The overall manuscript is interesting and presents the results well.
Author Response
Manuscript ID: sensors-1878271
Development of a GIS-Based Methodology for the Management of Stone Pavements Using Low-Cost Sensors
Answer to Reviewers’ Comments
First of all, we thank the Reviewers for their valuable comments and suggestions that helped us to improve the quality of our paper.
We revised the manuscript taking into account all the reviewers’ concerns.
Reviewer #2:
Reviewer #2:
The overall manuscript is interesting and presents the results well.
- 1 - not all arrows are fully visible.
Answer 2.1 The Reviewer is right. The authors modified Fig.1.
- Line 48-50 the language is not quite clear.
Answer 2.2 We thank the reviewer for the useful comment. We have modified the text at lines 47-49 (ex. 48-50):
“In recent years, the number of PMSs applied for urban systems has increased [2,26,27]. However, this trend has not involved stone pavements.”
